# GIS-Based Analysis of Volatile Organic Compounds in Bucheon, Korea, Using Mobile Laboratory and Proton-Transfer-Reaction Time-of-Flight Mass Spectrometry Methods

**DOI:** 10.3390/toxics12070511

**Published:** 2024-07-15

**Authors:** Minkyeong Kim, Daeho Kim, Jung-Young Seo, Duckshin Park

**Affiliations:** 1Railroad Test & Certification Division, Korea Railroad Research Institute (KRRI), Cheoldo Bangmulgwanro, Uiwang-si 16105, Republic of Korea; mkkim15@krri.re.kr; 2Department of Environemtnal Science and Ecological Engineering, Korea University, Seoul 02841, Republic of Korea; daehoya@korea.ac.kr; 3Department of Horticulture & Landscape Architecture, Andong Science College 189, Seoseon-gil, Seohu-myeon, Gyeongsangbuk-do, Andong-si 36616, Republic of Korea; jounseo@asc.ac.kr; 4New Transporation Innovation Research Center, Korea Railroad Research Institute (KRRI), Cheoldo Bangmulgwanro, Uiwang-si 16105, Republic of Korea

**Keywords:** geographic information system, hotspot analysis, mobile laboratory, spatiotemporal distribution characteristics, volatile organic compounds

## Abstract

Recently, volatile organic compounds (VOCs) have been shown to act as precursors of secondary organic particles that react with ultraviolet rays in the atmosphere and contribute to photochemical smog, global warming, odor, and human health risks, highlighting the importance of VOC management. In this study, we measured VOC concentrations in various contexts including industrial and residential areas of Bucheon, Korea, through mobile laboratory and proton-transfer-reaction time-of-flight mass spectrometry methods to determine winter VOC concentrations and visualized the data based on spatial information. Regional characteristics, temperature/humidity, atmospheric conditions, wind speed, traffic volume, etc., during the measurement period of the study site were comprehensively reviewed. For this purpose, global information system (GIS)-based air quality data and various environmental variables were comprehensively reviewed to assess spatial and temporal concentrations in three dimensions rather than in tables and graphs. Among VOCs, the levels of toluene, methanol, and n + i-butene were relatively high, with average concentrations of 48.3 ± 67.2, 34.4 ± 102.7, and 32.6 ± 57.7 ppb, respectively, at the end of the working day. The highest concentrations occurred near the Ojeong Industrial Complex. Mobile pollution sources are also a major driver of VOCs, highlighting the necessity of comprehensively reviewing traffic variables such as road level, estimated traffic volume, and average speed when identifying hotspots of air pollution. GIS-based visualization analysis techniques will improve the efficiency of air quality management.

## 1. Introduction

In Korea, the Clean Air Policy Support System identifies nine air pollutants (total suspended particulates, ultrafine particles [PM2.5], fine particles [PM10], sulfur oxides [SOx], nitrogen oxides [NOx], volatile organic compounds [VOCs], ammonia [NH3], carbon monoxide, and black carbon) originating from point, surface, and mobile pollution sources annually in Korea. According to the National Fine particles Information Center of the Ministry of Environment and based on calculated national air pollution emissions in 2021, the total emissions of VOCs were 1,002,810 tons, with Gyeonggi Province accounting for approximately 20% at 188,789 tons, the highest proportion [1]. In particular, based on emissions from 2016 to 2021, the implementation of the Comprehensive Fine Particles Management Plan and the Second Fine Particles Seasonal Management System resulted in decreased emissions of PM2.5, SOx, and NOx, whereas VOC and NH3 emissions increased. These increases were attributed to increased usage of paints for ships and construction.

Major anthropogenic sources of VOCs include automobiles, painting facilities, oil reservoirs, gas stations, the printing and ink industries, wastewater treatment plants, road paving, and agriculture. In particular, emissions related to the use of organic solvents, including the painting of automobiles and ships, account for 50% of total emissions, whereas emissions of automobile-related mobile pollutants account for 13.5% [2,3,4,5]. VOCs, air pollution-causing substances that deteriorate air quality, are hydrocarbon compounds that readily volatilize into the air in the forms of ozone and secondary aerosol precursors, contribute to climate change, worsen air quality, and pose risks to human health [6,7,8,9,10,11]. Among VOCs, benzene, formaldehyde, and 1,3-butadiene are classified as carcinogens and acetaldehyde, acrylonitrile, chloroform, and 1,2-dichloroethane are managed as specific air pollutants. VOCs in the human body can cause drowsiness, difficulty breathing, vomiting, nausea, abdominal cramps, unconsciousness, suffocation, lung cancer, leukemia, central nervous system disorders, asthma, and emphysema [12,13,14,15,16]. Accordingly, the importance of VOC management has recently been highlighted due to its impact on air pollution [17].

Bucheon is a megacity connected to Seoul and Incheon in Gyeonggi Province, where the combination of exhaust from automobiles and pollutants emitted from production processes within industrial complexes is worsening urban air quality. In a 2007 survey, Bucheon was found to have a VOC concentration of 71.916 ppb, the highest in Gyeonggi Province, due to its high level of industrial activities. However, in areas with diverse emission characteristics, investigating the emission sources can be challenging [18]. Five air pollution measurement stations are located in Bucheon (4 urban air quality monitoring systems [AQMS] and 1 roadside air quality monitoring system [RAQMS]), which obtain 10 types of air quality measurements in real-time. However, although these fixed measurement networks can analyze the impact of air pollution on a single receptor at a given location, they have limitations in examining broader impacts on the surrounding area and accurately identifying mixed pollutants. Recently, mobile laboratories (MLs) have been used to supplement this network. Many local governments in Korea are expanding the use of ML vehicles and drones to monitor air quality in areas without existing monitoring stations. However, the collected data are often not effectively visualized [19].

Many previous studies have been conducted using ML methods to examine the concentrations and spatial distributions of VOCs in urban areas and around petrochemical complexes and publishing complexes [6,12,15,18,20,21]. However, to date, no comprehensive air pollution management strategy has been implemented using geographic information system (GIS)-based data visualization or digital twin-based models and environmental data, which have been recently developed.

This study is the first to visualize the relationship between ML-based time-series data and environmental factors that can affect VOCs using GIS-based spatiotemporal analysis. Bucheon comprises residential, industrial, and industrial settings and this study will support policy decision-making by identifying hotspots and areas requiring intensive management through visualization of the factors that can affect VOCs as well as their spatial and temporal patterns. This study also aims to demonstrate that GIS-based air quality management is essential in some contexts

## 2. Materials and Methods

### 2.1. Research Site

As of 2020, Bucheon, Korea, has an area of 53.4 km^2^, a population of 831,701, and a population density of 15,575 people/km^2^. Bucheon has the highest population density in Gyeonggi Province and contains a high concentration of residential, industrial, and industrial facilities. Bucheon is located between Seoul and Incheon, adjacent to the Gyeongin Expressway and the Capital Region First Ring Expressway, with a traffic volume of 3.3 million people per day, which is very high relative to its population. The target site for this study was adjacent to the Gyeongin Expressway and First Ring Expressway in the metropolitan Bucheon area and the survey route was selected near industrial areas and major roads that may impact VOCs (Figure 1). The weather conditions during the measurement period on 6 December were an average temperature of 6.3 °C, a maximum temperature of 10.9 °C, and a minimum temperature of 1.7 °C; the average humidity was 51.4% and wind speed was 1.7 m/s. On 7 December, the average temperature was 7.0 ℃, the highest temperature was 10.8 °C, and the lowest temperature was 3.5 °C; the average humidity was 56.5% and the wind speed was 2.9 m/s. And there was haze on both days. Haze occurred frequently during December. In this study, the air quality monitoring system (AQMS) is a measurement station in Nae-dong and the Songnae-daero roadside air quality monitoring system (RAQMS) is a measurement station in Songnae-daero. The air quality monitoring system (AQMS) measured PM10 (85.1 ± 21.5 µg/m^3^) and PM2.5 (52.7 ± 15.1 µg/m^3^) concentrations. December was selected because the concentration of fine particles and VOCs is high in winter. In particular, the measurement period was selected as a day without rainfall and, based on this, the winter trend was identified. In this study, the concentration in the air in the form of stationary measurements and the concentration on the road in the form of mobile measurements were compared over two days.

The subject of this study is Bucheon City, where there is a road that is intensively managed due to high traffic volume and the average speed is as follows. The average speed of the previous day on the Gyeongin Expressway was 60 to 74 km/h and 33 to 47 km/h at 8 a.m. and 8 a.m., respectively, peak times for commuting and returning from work. In the case of the Gyeongin Expressway connection road, the average speed was 16 to 37 km/h as of the previous day. The average speed of the previous day on the 1st Ring Expressway in the metropolitan area was 43 to 62 km/h and similar speeds were observed including commuting and finishing hours. In the case of the connecting road of the 1st Ring Expressway in the metropolitan area, the average speed was 31 to 53 km/h the previous day and similar speeds were observed including commuting and leaving work hours. In the case of Songnae-daero, the average speed of the previous day was 27 to 57 km/h and, in the case of Gilju-ro, the average speed of the previous day was 17 to 49 km/h. In the case of highways, the average speed is rapidly decreasing on the connecting roads and traffic congestion sections are lengthening accordingly.

### 2.2. Experimental Methods

#### 2.2.1. Laboratory Equipment

We used an ML to obtain location information and measure air pollutants (Figure 2). MLs enable real-time measurement of the spatiotemporal distributions and concentrations of air pollutants within a target area. However, the time delay between the flow of pollutants into the inlet and measurement, non-uniform absorption due to the physical characteristics of different pollutants, and separation from Global Positioning System (GPS, https://www.gps.gov/, accessed on 3 July 2024) data due to equipment sensitivity may affect the results [18,22,23,24,25,26].

In this study, proton-transfer-reaction time-of-flight mass spectrometry (PTR-ToF-MS; Ionicon Analytik, Innsbruck, Austria), which can acquire and analyze data on individual VOCs in real time, was used as the main measurement technology. PTR-ToF-MS(Ionicon Analytik, Innsbruck, Austria) analyzes VOCs through the proton transfer reaction; compared to gas chromatography-mass spectrometry (GC-MS), this method has the advantage of analyzing the concentrations of individual VOCs in the air in seconds without sample collection or pretreatment. In general, when using GC-MS for VOC analysis, significant time is required for pre-processing prior to analysis. In contrast, PTR ToF-MS analysis is completed within 1 s, without pre-processing. This method presents measurement results in practical units, making it highly suitable for mobile measurement [27,28,29,30,31,32,33]. The sample inlet is located at the top of the vehicle and is constructed at a height of approximately 3 m above the ground to prevent rainwater from entering. Air samples are transferred to the PTR-ToF-MS through a suction pump installed inside the vehicle and a portion of the transferred sample (15 mL/min) is introduced into the PTR-ToF-MS through the connector of the sample inlet. The sample transfer tube is made of Teflon and is about 3 m long. In addition, in order to minimize loss due to adsorption within the pipe during the transfer process, a heating wire was installed to enable heating up to 120 °C. In addition, before measurement, a one-day calibration was performed using standard samples and high-purity clean air and a calibration curve for each substance was created to check whether the equipment was operating normally. We used a GPS device to determine the spatial coordinates of each measurement collected during vehicle movement to prevent measurement errors due to time differences among measurement devices (Table 1). An intake port was installed on the roof of the vehicle to collect air samples while driving; it was designed to collect pollutants at a constant velocity. At vehicle speeds of 0–70 km/h, we confirmed the average speed to have a rate of change of less than 0.8–2.0%; therefore, measurements in traffic were conducted within this speed range as much as possible [34].

The ML vehicle was remodeled after approval of the structural changes. The vehicle was equipped with a PTR-ToF-MS instrument and a generator for measuring individual VOCs. At vehicle speeds of 0–70 km/h, we confirmed that vehicle speed caused changes of less than 0.8–2.0%; therefore, measurements in traffic were conducted within this speed range whenever possible [34].

#### 2.2.2. Target Substances

In this study, 16 major pollutants were selected; information about each substance is summarized in Table 2. For each target material, calibration was performed using standard samples and high-purity clean air prior to measurement; the equipment was confirmed to be operating normally through fixed measurements.

#### 2.2.3. Measurement Procedure

Mobile measurements were taken along a predetermined route, which started and ended at the Bucheon Technopark in Yakdae-dong; prior to operation, the equipment was inspected and the battery was charged. Beginning at the Technopark, the route passed through Gyenam Park and the Eugene ready-mixed concrete complex and near to Songnae-daero, Gilju-ro, Gyenam-ro, and the Gyeongin Expressway. The route was 19.3 km in length and took about 1 h to complete.

An analysis of seasonal concentrations showed that VOC concentrations were highest in winter; therefore, this study was conducted on 6 and 7 December. Seven sets of moving measurements (five on 6 December and two on 7 December) were conducted during regular commuting and working hours. The driving speed was maintained at approximately 20–30 km/h.

#### 2.2.4. Geographic Information System (GIS) Analysis

We used the open-source software QGIS ver. 3.32 to analyze the spatiotemporal distribution of air pollution in Bucheon. GIS analysis was conducted using the GPS-derived coordinates of the ML along with time and concentration data for the measured pollutants.

Most VOCs occur in gaseous form and organic solvent use such as painting accounts for the largest portion of VOC emissions. Another important emission source is automobile-related pollutants released during travel. Accordingly, data such as road information, vehicle speed (average speed in congested traffic and maximum speed), and traffic volume were collected and included as variables in our analysis. For traffic volume, we used publicly available data from the Korea Transport Institute national transport database, View-T (Figure 3). This site includes data from 1 lane to a maximum of 11 lanes, which is indicated.

## 3. Results and Discussion

### 3.1. Concentration Distribution of VOCs

The path of the ML included a mixture of residential and industrial areas and Table 3 presents the fixed observation data, excluding moving observations. In this study, stationary measurements were performed from the ML at Bucheon Techno Park in Yakdae-dong. In particular, the concentration of VOCs on the roadside in the industrial area was confirmed by examining the concentration in the air in the form of fixed measurements and the concentration on the road in the form of moving measurements.

Among VOCs, benzene is the subject of environmental standards, with an annual average of 1.5 ppb as the national standard. And, the concentrations of VOCs measured during the fixed measurement period decreased in the order of methanol > toluene > n + i-butene. Overall, methanol accounted for 17.0%, toluene for 16.5%, and n + i-butene for 12.3% of total VOCs (Table 3). Developed countries such as the United States, United Kingdom, and Japan have set environmental standards for VOCs since 1990 [35] and Korea has introduced annual average standards for benzene.

The main areas of research related to VOCs are around petrochemical complexes and publishing complexes and there is little research on VOCs’ concentration targeting major roadsides in urban areas. In addition, even in the case of studies that measured VOC concentrations by mobile measurement, results on the trends of VOCs were derived based on the results of a minimum of 2 days and a maximum of 5 days, depending on the limitations of using a mobile laboratory. A study [5] that looked at the concentration of VOCs during the four seasons in the area around the petrochemical complex found that winter actually showed the highest concentration at 18.6 ppb, summer at 12.0 ppb, and spring and fall at 10.5 ppb. Based on this, this study was targeted at winter. A study [6] that measured the concentration around the publishing complex targeted September. As a result of examining the concentration ratio for each individual substance, it was confirmed that toluene (22.51 ± 8.29%) and methanol (19.01 ± 3.73%) appeared in the ratio and, as in this study, it was confirmed that the concentration of the two substances appeared at a high ratio. In a study [12] that measured the concentration around a petrochemical complex, the target was June; toluene and benzene were detected at relatively high levels compared to other substances and had the characteristic of showing peaks around 12 o’clock and 19 o’clock. In addition to the pollutants emitted from the industrial complex, it was confirmed that the country is also affected by many pollutants emitted from cars commuting to or delivering goods to the industrial complex. Through research results derived from movement measurements, it was confirmed that the concentration of VOCs on roadsides with high traffic volume was higher than that in industrial complexes. Based on this, this study selected a target area with a very high population density and a very high daily traffic volume and examined the effect of road emissions on VOC concentration.

This study did not target specific complexes and instead examined the spatial and temporal distributions of various VOC types in the city center using ML. The VOC concentrations measured during the movement of the ML decreased in the order of toluene > methanol > n + i-butene. When measured statically, methanol had a higher concentration than toluene but, when measured from the moving ML, the opposite pattern was observed. Comparing moving and stationary measurements, the concentrations of Methanol, Acetaldehyde, n + i-butene, Acetic acid, i + n-Pentane, Benzene, Toluene, and Ethanol + xylene are relatively higher during moving measurements than during stationary measurements. During the actual measurement period, the section in question was an urbanized area as a regional variable and it was a section with very high traffic volume on weekdays and weekends. Also, the wind speed was not strong and there was no effect of rainfall. Looking at the wind direction on 6 December, it was a southeast wind from 8 a.m. to 11 a.m. and, after that, it was a northwest wind. Looking at the wind direction on 7 December, it was a northeast wind from 8 a.m. to noon and, after that, it was a northwest wind. However, since the wind speed was below 1.0 m/s, the influence of wind direction can be considered low.

However, as of 2021, Bucheon City’s annual average PM10 concentration is 48 µg/m^3^ and PM2.5 concentration is 24 µg/m^3^; while reflecting the seasonal characteristics of winter, it can be confirmed that the annual average concentration is more than twice as high. there is. In the case of the Nae-dong air quality monitoring system (AQMS) station (Figure 1), during the period, the PM10 concentration was 85.1 ± 21.5 µg/m^3^ and the PM2.5 concentration was 52.7 ± 15.1 µg/m^3^. And, in the case of the Songnae-daero roadside air quality monitoring system (RAQMS) station (Figure 1), PM10 concentration was found to be 63.5 ± 12.6 µg/m^3^ and PM2.5 concentration was found to be 38.9 ± 8.3 µg/m^3^. During the two days measured by the mobile measurement vehicle, the concentration of fine dust was also measured and, at the first measurement on 6 December, the PM10 concentration was 79.6 ± 19.7 μg/m^3^ and the PM2.5 concentration was 62.7 ± 13.0 μg/m^3^ and, at the time of the fifth measurement on 6 December, the PM10 concentration was relatively high at 91.8 ± 17.6 μg/m^3^ and the PM2.5 concentration was 59.2 ± 8.8 μg/m^3^. The NO_2_ concentration was highest during the morning and evening rush hour (0.08 ± 0.03 ppm). Fine particles are most often caused by automobile fuels and, in the case of PM2.5, it is generated secondarily by reacting under certain conditions in the atmosphere, such as NOx and VOCs. In this study, fine particle concentration, wind speed, traffic volume, temperature/humidity, etc., were considered and the seasonal characteristics of winter and the high traffic volume in the area were derived as major variables. Based on this, we attempted to examine the effects of VOCs on vehicle operation on the road.

The concentrations of each VOC type at each measurement period are presented in Table 4. Most studies to date have shown that VOC concentrations are high in winter; this study examined the temporal and spatial distributions of VOCs during the winter period.

The average concentration values for 16 individual VOCs on the first and second days were compared (Table 5). The differences between the 16 individual substances showed a similar trend and, in the case of methanol and toluene, the average concentration was found to differ by more than 5 ppb. However, the rankings and VOC ratios among the 16 substances were derived as showing similar trends between stationary and moving measurements.

### 3.2. Spatiotemporal Pattern of VOC Concentration Considering Environmental Variables

Local air management for VOCs requires the identification of substances with the greatest environmental and health importance. In this study, among the 16 substances measured, 3 substances with high concentrations and 1 substance that met the applicable atmospheric standard were included in the spatial analysis (Figure 4, Figure 5, Figure 6 and Figure 7). In addition, we reviewed the seventh result in which acetonitrile, an indicator of biomass combustion, had a relatively high measured value. Measurements were made in 1-s increments during the movement of the ML. Because the measurement points were too densely spaced to visualize the observations in 1-s increments, the results were displayed using GIS after averaging over 10-s increments.

As a result of examining the concentration of VOCs and the influence of other environmental factors during the measurement period (Table 6), the concentrations of PM10, PM2.5, and NOx in the fifth measurement period, which was evening, were relatively high. In addition, in this study, the results of comparing four substances with high concentrations among the 16 VOC substances showed that the concentrations were relatively high in the fifth measurement period; a spatial hotspot analysis was performed based on this.

In this study, the concentration during the fifth measurement time was determined to be the highest and, based on this, the correlation coefficient between the four substances presented above and the environmental variables was derived (Table 7). As a result of examining the correlation, among the observed items for Toluene, Methanol, n + i Butene, and Benzene, which are the main target substances in this study, the parts with relatively high correlation were examined under limited conditions.

Toluene, which accounted for a large proportion of VOCs in this study, is a representative VOC emitted into the atmosphere from anthropogenic activities. Toluene decomposes in the air through a reaction with optically generated OH radicals. The concentration of toluene increases in the presence of vehicle emissions and exhibits large increases in areas of direct use, such as industrial processes and solvent treatments. In particular, the proportion of toluene is high when the contribution from automobile exhaust is large [36]. We found that toluene concentrations were relatively high when the ML traveled the Gyeongin National Road and passed through the Ojeong Ready-Mixed Concrete Industrial Complex and that they were particularly elevated during off-peak hours (Figure 4a).

The highest average toluene concentration was 48.3 ± 67.2 ppb, observed during the period when people were leaving work. However, the spatial and temporal distribution of toluene during mobile measurements, in contrast to the average value, had a maximum of 640.64 ppb and a minimum of 1.93 ppb.

The correlation between toluene and environmental variables was examined and the variable with the highest correlation was PM2.5, with a correlation coefficient of 0.47 (Table 7). The spatial and temporal distribution of this was examined based on GIS (Figure 4). As a result, it was confirmed that the toluene concentration was high when passing through roads and intersections leading to the Ojeong Ready-Mixed Industrial Complex, similar to the PM 2.5 concentration. Visual assessment of areas with low toluene levels allowed us to identify the regional characteristics that cause variations in the toluene concentration.

Methanol had the second-highest average concentration among VOCs. Methanol is particularly toxic to humans and can cause destruction of the optic nerve and permanent visual impairment. Like toluene, the average concentration of methanol was highest while most people were leaving work, at 34.4 ± 102.7 ppb. However, assessing the spatial and temporal distribution using GIS with moving measurements, in contrast to simply using the average value, showed that the highest concentration reached 1242.43 ppb and the lowest was 7.01 ppb (Figure 5).

As a result of examining the correlation between methanol and environmental variables, the overall correlation was not high but the NOx results were examined as a variable with a relatively high correlation (Table 7). The spatial and temporal distribution of this was examined based on GIS (Figure 5). As a result, it can be seen that the concentration is relatively high on the road, leading to the Ojeong Readymix Industrial Complex and the Gyeongin Expressway area. Thus, we were able to visually determine which regions had the highest and lowest concentrations. Management measures for VOCs must be established based on analysis of the results according to regional characteristics rather than simply confirming that the average concentration is high. Therefore, we examined the spatial and temporal distributions of VOC concentrations based on GIS analysis.

The substance with the third-highest concentration was n + i-butene. A relatively high average concentration was observed during work hours, at 32.6 ± 57.7 ppb. However, assessing the spatial and temporal distribution using GIS with moving measurements, in contrast to simply using the average value, showed that the highest concentration reached 670.98 ppb and the lowest was 3.91 ppb (Figure 6).

As a result of examining the correlation between n + i-butene and environmental variables, the overall correlation was not high but the relatively high variable was PM10, with a correlation coefficient of 0.17 (Table 7). The spatial and temporal distribution of this was examined based on GIS (Figure 6). As a result, it was confirmed that the n + i-butene concentration was similar to the PM10 concentration on the roads and intersections leading to the Ojeong Readymix Industrial Complex and the Gyeongin Expressway area and when passing through intersections. Thus, results are available for each point along the ML route, allowing us to determine which regions had the highest and lowest concentrations.

In addition to the three high-concentration substances noted above, benzene, which is subject to an environmental standard, was also measured. The air quality standard for benzene is an annual average concentration of 1.5 ppb and the highest average concentration measured in this study was 5.0 ± 10.3 ppb. The average concentrations of the seven measurement periods consistently exceeded the air quality standard. Based on the spatial and temporal distribution obtained using GIS, we visually confirmed that the lowest concentration was 0.54 ppb and the highest concentration reached 77.92 ppb (Figure 7).

As a result of examining the correlation between benzene and environmental variables, the variable with the highest correlation was PM2.5, with a correlation coefficient of 0.55 (Table 7). The spatial and temporal distribution of this was examined based on GIS (Figure 7). As a result, it was confirmed that the benzene concentration was similar to the PM2.5 concentration in the area of the road leading to the Ojeong Readymix Industrial Complex and the Gyeongin Expressway. Benzene is highly volatile and disperses very quickly in the air, leading to exposure via the respiratory tract. Benzene is a carcinogen, and exposure to high concentrations can cause blood cancers such as leukemia. Additionally, short-term exposure to high benzene concentrations can affect the nervous system, causing dizziness, headaches, confusion, unconsciousness, and eye and skin irritation. Long-term exposure reduces white blood cell and red blood cell levels, causing anemia and lowering immunity. Accordingly, rather than presenting benzene levels in a simple table, management measures should be developed based on visualization of regional concentrations, allowing for immediate confirmation [37].

Relatively high concentrations of the four substances presented were observed in the Ojeong Readymix Industrial Complex section, the Gyeongin Expressway entrance section, and the intersection, where the VOC concentration tended to be equally high. In addition, when managing air quality through a review of environmental variables with high correlation coefficients, special management is required for areas with high concentration. Especially, according to domestic air quality standards, the annual average for benzene is approximately 1.5 ppb or less and the annual average standard for PM2.5 is 25 μg/m^3^. However, as the average concentration is more than double during some periods of winter in the region, special management is required during that period. Currently, the local government has installed sensors in various sections and configured them as a platform to establish a system that sounds an alarm in areas where the concentration is increasing.

Acetonitrile, an indicator of biomass combustion, showed a relatively high concentration of 2.8 ± 6.0 ppb in the seventh measurement out of a total of seven results. Acetonitrile is a colorless compound that smells like ether and exposure to high concentrations of acetonitrile can cause breathing difficulties, nausea, vomiting, convulsions, and coma. In particular, as the route covered by this study includes many residential areas in addition to industrial areas, the impact by time was analyzed based on GIS (Figure 8). As a result of examining the correlation between acetonitrile substances and other environmental items, PM10 was derived as 0.25, PM2.5 as 0.11, NO as −0.03, NO_2_ as 0.08, and NOx as −0.01. As the correlation with PM10 was relatively high among the items, the correlation results between acetonitrile and PM10 were reviewed based on GIS. Acetonitrile and PM10 concentrations were found to be similarly high in the road section and intersection area, leading to the Ojeong Readymix Industrial Complex and Gyeongin Expressway. It passes through residential areas such as Naedong and Dodang-dong in Bucheon City, and the concentration of acetonitrile, which is an indicator of biomass combustion, is above a certain level in the area, so management that reflects its characteristics is necessary.

### 3.3. Spatiotemporal Pattern of VOC Concentration Considering Traffic Data

In Bucheon, the Gyeongin Expressway passes through Songnae-dong, Simgokbon-dong, Sosabon-dong, and Goean-dong. During ML measurement, the concentration of VOCs generated on the road was measured while traveling along a route that included the entirety of the residential and industrial areas in the study region. In this study, hotspot analysis was performed on three VOCs found in high concentrations and one substance covered by air quality standards based on average speed and expected traffic volume. In other studies [15,18,20], the concentration of VOCs on nearby roads was found to be high due to the movement of vehicles entering the industrial complex; the impact of mobile pollutants on the concentration of VOCs was also found to be high.

As a result of examining the correlation between the four VOCs presented above and average speed and expected traffic volume, the correlation coefficient between Toluen and Benzene was found to be 0.14 in the case of average speed. As a result of examining the correlation between the expected traffic volume and the four substances, it was concluded that the overall correlation was low (Table 8).

As a result of examining the correlation between average speed and expected traffic volume, the correlation coefficient was found to be low. However, through spatial and temporal distribution results based on actual data and GIS, the impact of individual VOC substance concentrations on average speed and expected traffic volume for each road section was visualized and reviewed.

As automobile-related mobile pollutant emissions are a major source of VOCs, we investigated the effects of traffic volume and average travel speed on toluene concentrations. As the average concentration (48.3 ± 67.2 ppb) was high in the evening, hotspots were identified through analysis of the spatial pattern of toluene concentrations using evening data. We identified hotspots of toluene levels during rush hour in congested road sections that had average traffic speeds ranging from 0 to 30 km/h. The maximum toluene concentration during the off-hours was very high, at 521.64 ppb. A comparison of the estimated traffic volume and toluene concentration confirmed that traffic volume increased to 1253–1895 vehicles/day where the toluene concentration was high (Naedong and Gyeongin-ro intersection) (Figure 9). In the case of connecting roads, delays occurred, resulting in a decrease in average speed and a tendency for traffic volume to increase. The proportion of toluene was high when the contribution of automobile exhaust gases was high, emphasizing the importance of pollution mitigation measures that account for traffic volume and average traffic speed.

For methanol, which had the second-highest concentration, we investigated the effects of traffic volume and average travel speed on its concentration. As the average concentration (34.4 ± 102.7 ppb) was high in the evening, hotspots were identified based on the spatial pattern of toluene concentration during the evening. We identified hotspots of methanol during rush hour in congested road sections that had average traffic speeds ranging from 0 to 30 km/h. In particular, frequent movement of ready-mix concrete vehicles toward the Ojeong Industrial Complex was observed in these areas. A prior study [12] similarly confirmed that the concentration was high due to vehicles frequently passing around industrial complexes. A comparison of the estimated traffic volume and methanol concentration confirmed that traffic volume increased to 1253–1895 vehicles/day in areas with high methanol concentrations (Ojeong Industrial Complex) (Figure 10). To manage the concentration of VOCs among automobile-related pollutants, comprehensively considering factors such as traffic volume is essential.

For n + i-butene, which had the third-highest concentration, we investigated the effects of traffic volume and average travel speed on its concentration. As the average concentration (32.6 ± 57.7 ppb) was high in the evening, hotspots were identified through analysis of the spatial pattern of n + i-butene concentrations in evening data. We identified hotspots of n + i-butene during rush hour on congested road sections with average traffic speeds ranging from 0 to 30 km/h. Movement of ready-mix concrete vehicles in the direction of Ojeong Industrial Complex was extensive and the concentration was high at the Naedong-Gyeongin-ro intersection. Comparing the estimated traffic volume and n + i-butene concentration confirmed that the maximum traffic volume reached 1253–1895 vehicles/day (Figure 11).

Benzene is a VOC included in air quality standards and is generally considered a marker of automobile exhaust gas and evaporative emissions. Therefore, the effects of estimated traffic volume and average travel speed in a road section on benzene concentration were investigated. Hotspots were identified based on the spatial pattern of benzene concentrations using data with a relatively high average concentration (5.0 ± 10.3 ppb) collected during the study period. We identified hotspots of benzene during rush hour in busy road sections with average traffic speeds of 0 to 30 km/h. The Naedong-Gyeongin-ro intersection and the Ojeong Industrial Complex area had benzene concentrations above 1.5 ppb, which is the annual average standard for air quality. Comparing the estimated traffic volume and benzene concentration confirmed that traffic volume increased from 799 to 1323 vehicles per day at the Naedong–Gyeongin-ro intersection and the Ojeong Industrial Complex, where benzene concentrations were high. In particular, as air quality standards are being proposed for benzene, management measures should be formulated using GIS-based spatial and temporal distribution data (Figure 12).

Our findings highlight the need to target and continuously manage road sections with high concentrations of VOCs that are harmful to humans. Studies to date have generally examined only the ratio and average concentration of each VOC type. Also, while other studies mainly focused on the surrounding areas such as petrochemical complexes and publishing complexes, this study can be seen as targeting major roads in urban areas with high traffic and population density. In the other research cases, the concentration of VOCs on the roadside was found to be high due to the movement of vehicles entering industrial complexes; the influence of mobile pollutants on the concentration of VOCs was also found to be high. Based on this, a GIS-based analysis was performed to examine the impact of VOC concentration on traffic volume and average speed. Pollution emitted by automobiles is a source of VOCs, making it necessary to identify pollution hotspots based on spatiotemporal analysis of traffic-related variables and to reflect those hotspots in air management policies. In this study, the concentration of each VOC type was determined and substances with high concentrations were visualized and reviewed. When the estimated traffic volume and average speed are included in the same category, they show similar concentration patterns. However, in some cases, even in the same category, the concentrations of individual VOCs were found to be different, such as at intersections of highways and general roads or in areas with dense industrial and commercial complexes. In future research, comprehensively reviewing various factors such as regional factors, traffic variables, and environmental variables to visualize GIS-based air quality management measures will be essential. Many local governments are currently using MLs to collect air quality data in real-time; however, simply considering the average values limits the identification of the source area and the establishment of efficient management measures. To address this limitation, GIS-based air quality management plans must be established.

## 4. Conclusions

Bucheon has the highest population density among the regions of Gyeonggi Province and contains industrial facilities and complexes in its downtown area. For these reasons, VOC types become mixed and are discharged at high concentrations. In general, determining the level of pollution according to the discharge characteristics of pollutants is important but most surveys of regional concentration levels use static process testers and only representative concentrations for AQMS and RAQMS can be obtained. In particular, VOCs are highly volatile, which allows them to be easily diluted and diffused when released into the atmosphere. Accordingly, to establish management measures for this area, confirming the spatial and temporal distribution of VOCs in the target area and developing air quality management measures based on information about the spread of those substances are essential. Therefore, this study used an ML to collect data for analysis using PTR-ToF-MS, which supports real-time measurement in seconds along routes passing through residential and industrial areas.

This study was conducted in winter (December) when VOC concentrations are seasonally high. Bucheon, the target area, is a city in Gyeonggi Province with a total daily average traffic volume of 20,000 vehicles as of 2019 and changes in the concentration of VOCs associated with mobile pollution sources were examined. We determined the concentrations of individual VOCs using PTR-ToF-MS, which can acquire and analyze ML and VOC data in real-time.

We confirmed that the average concentration of toluene, methanol, and n + i-butene were 48.3 ± 67.2, 34.4 ± 102.7 ppb, and 32.6 ± 57.7 ppb, respectively. Temporally, the concentration was high during work hours and spatially, the concentration was high on special management roads with high traffic and roads around the Ojeong Industrial Complex. Based on this, a GIS-based spatiotemporal analysis was performed to examine the impact of traffic volume and average speed on VOC concentration in this area. In the case of this region, conditions such as regional factors, temperature and humidity, precipitation, and wind speed did not appear to be major influencing factors, so the analysis was conducted based on the following: environmental variables (PM and NOx), traffic data, GIS-based road level, visualization of traffic volume (average and in congestion), and pollution level data overlaid with predicted traffic volume allowed the identification of pollution hotspots. In this study, as a result of examining the correlation between traffic variables and environmental variables in the case of VOCs, it was confirmed that environmental variables such as PM10, PM2.5, and NOx had a relatively high influence on the concentration of specific substances. Based on this, it was confirmed that hotspot analysis and prediction would be possible.

In addition, we used an ML to measure real-time VOC concentrations and individual VOC characteristics to identify high-concentration areas. Among VOCs, only benzene is currently reviewed in terms of air quality standards but other VOCs require similar analysis. It is necessary to analyze the impact of each substance according to variables such as environment and traffic. Previously, only the ratio and average concentration of each VOC type were assessed; therefore, this study is significant in that it derived results through the visualization of spatial and temporal pollutant distributions by overlaying maps of time-series data, environmental variables (PM and NOx), and traffic variables. Many local governments are currently using MLs to measure road pollution; to make accurate decisions based on the resulting scientific data, GIS-based analysis of air quality measurements collected in the field is needed. Through such analysis, it will be possible to identify areas with high levels of pollution and contribute to source determination based on the characteristics of highly volatile VOCs. In the case of major urban centers, various types of roads such as highways and general roads are connected and it is necessary to examine changes in VOC concentration due to emission sources on the roads. In the future, we plan to expand this study and conduct additional analysis and comparison of pollutants in various regions over a long period of time.

## Figures and Tables

**Figure 1 toxics-12-00511-f001:**
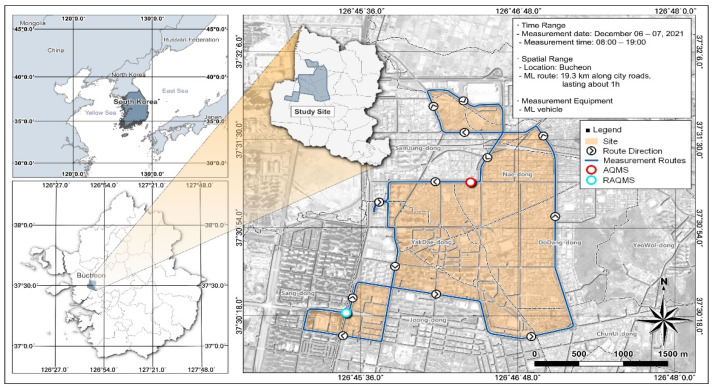
Location of the study site and mobile laboratory (ML) route.

**Figure 2 toxics-12-00511-f002:**
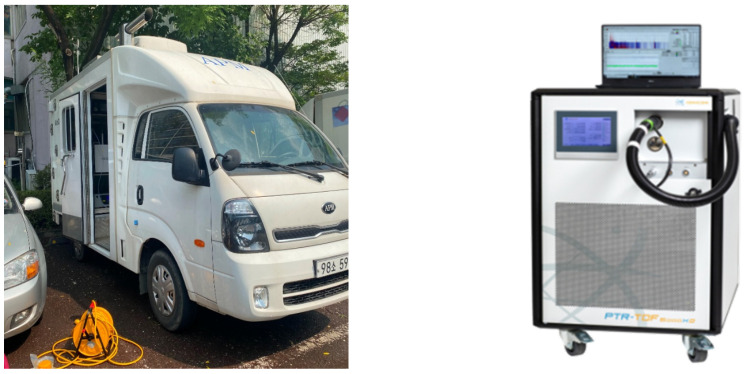
Air pollution ML vehicle (**left**) and proton-transfer-reaction time-of-flight mass spectrometry measurement instrument (**right**).

**Figure 3 toxics-12-00511-f003:**
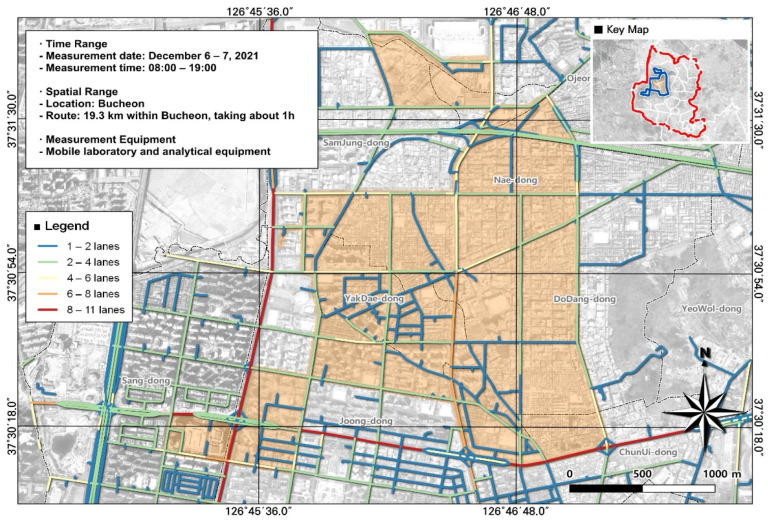
Numbers of lanes in roads within the study area.

**Figure 4 toxics-12-00511-f004:**
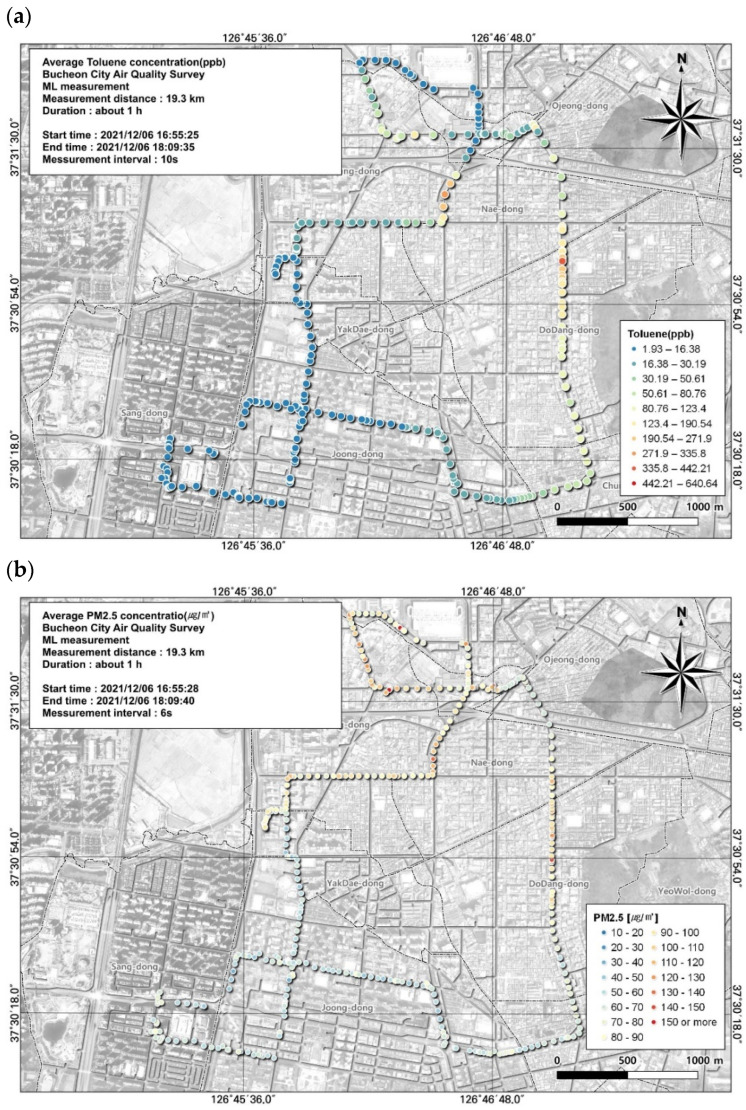
Spatial comparison of VOC (toluene) and PM2.5 concentrations measured from the ML (6 December, fifth run): (**a**) VOC (toluene) and (**b**) PM2.5.

**Figure 5 toxics-12-00511-f005:**
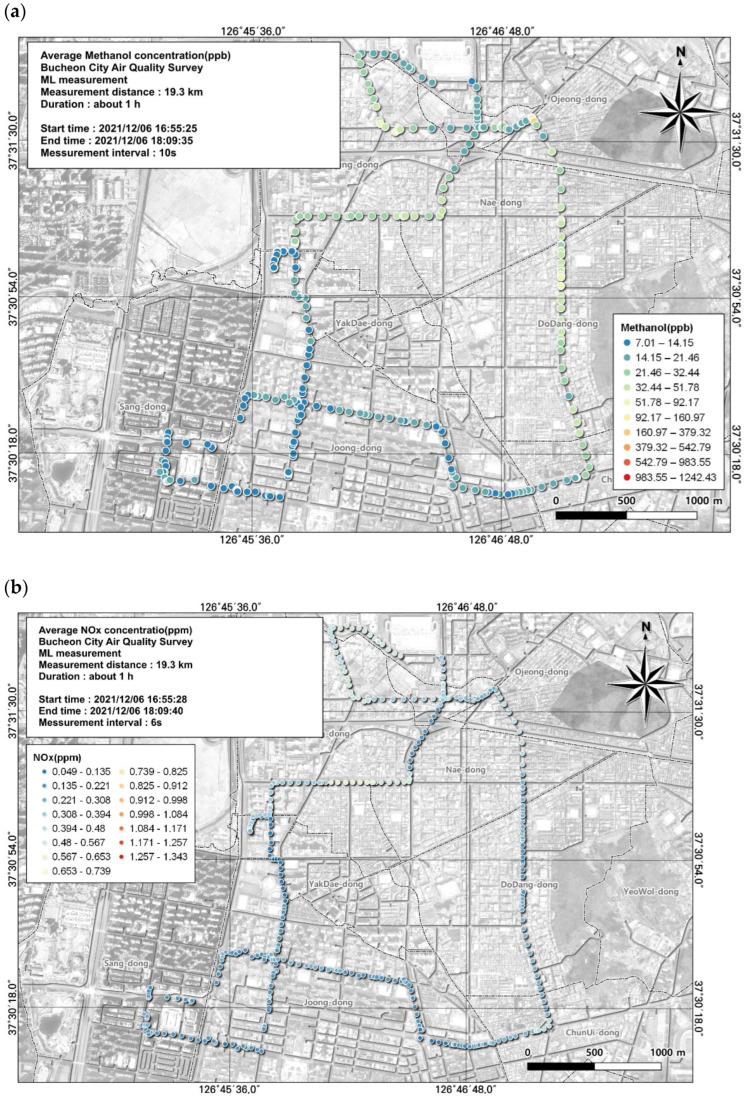
Spatial comparison of VOC (methanol) and NOx concentrations measured from the ML (6 December, fifth run): (**a**) VOC (toluene) and (**b**) NOx.

**Figure 6 toxics-12-00511-f006:**
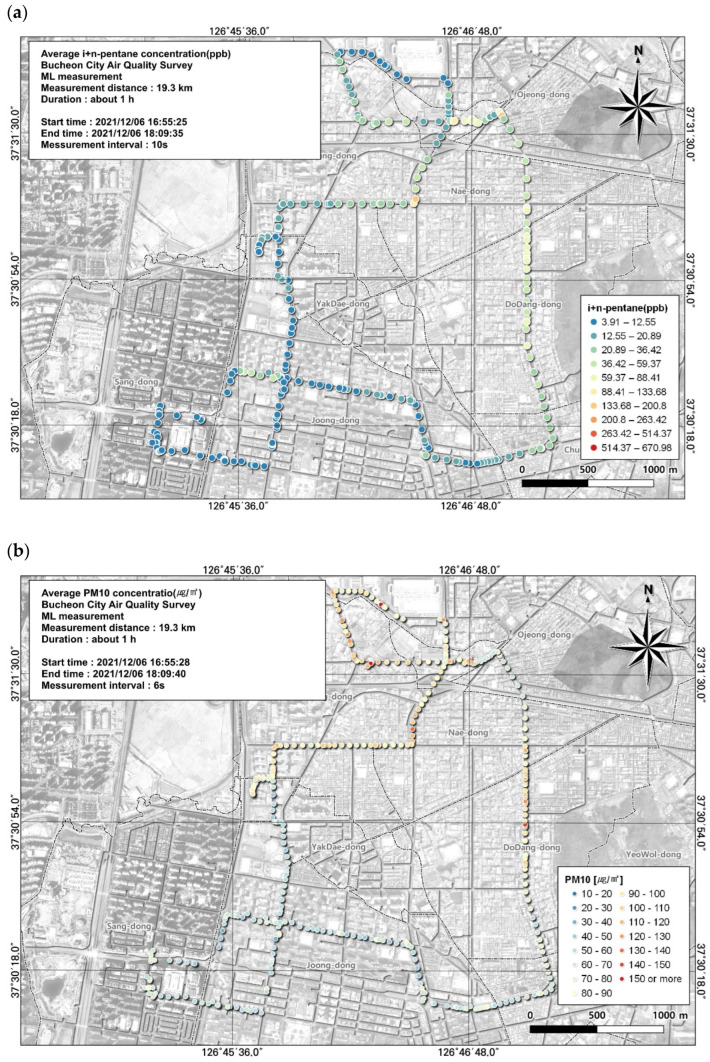
Spatial comparison of VOC (n + i-butene) and PM10 concentrations measured from the ML (6 December, fifth run): (**a**) VOC (toluene) and (**b**) PM10.

**Figure 7 toxics-12-00511-f007:**
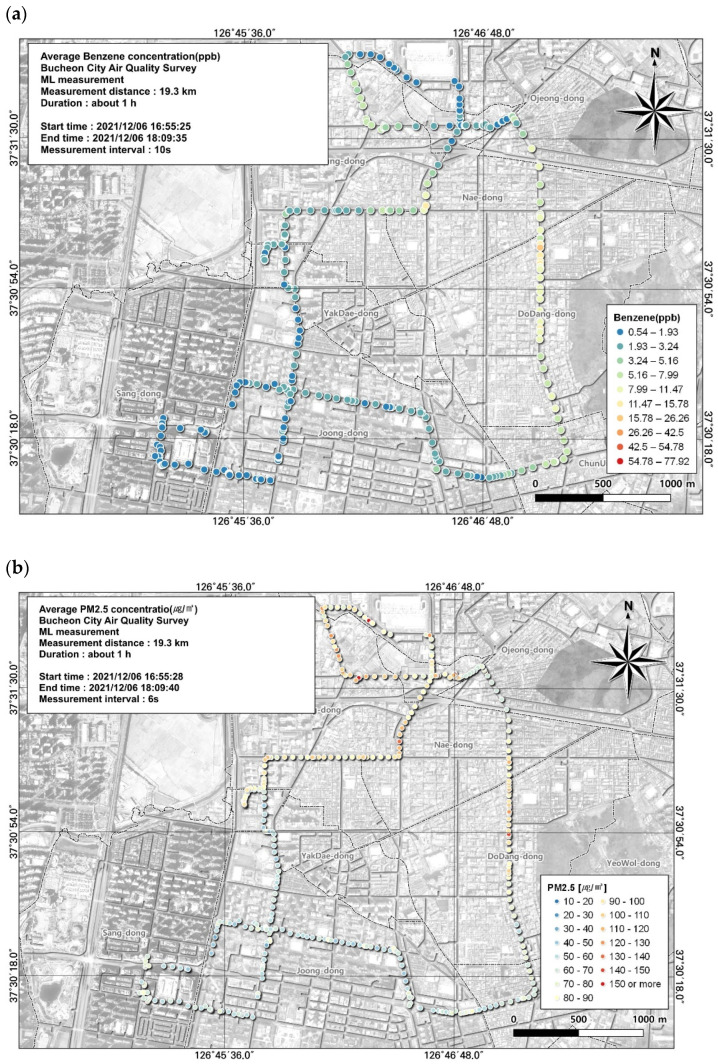
Spatial comparison of VOC (benzene) and PM2.5 concentrations measured from the ML (6 December, fifth run): (**a**) VOC (toluene) and (**b**) PM2.5.

**Figure 8 toxics-12-00511-f008:**
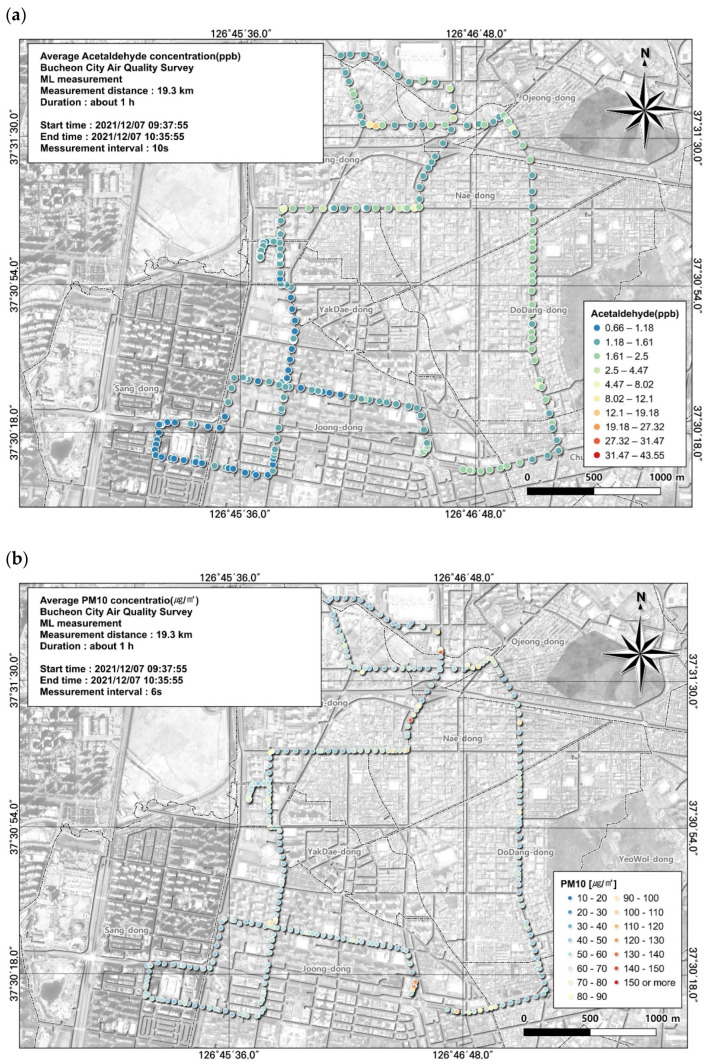
Spatial comparison of VOC (acetonitrile) and PM10 concentrations measured from the ML (7 December, 7th run): (**a**) VOC (toluene) and (**b**) PM10.

**Figure 9 toxics-12-00511-f009:**
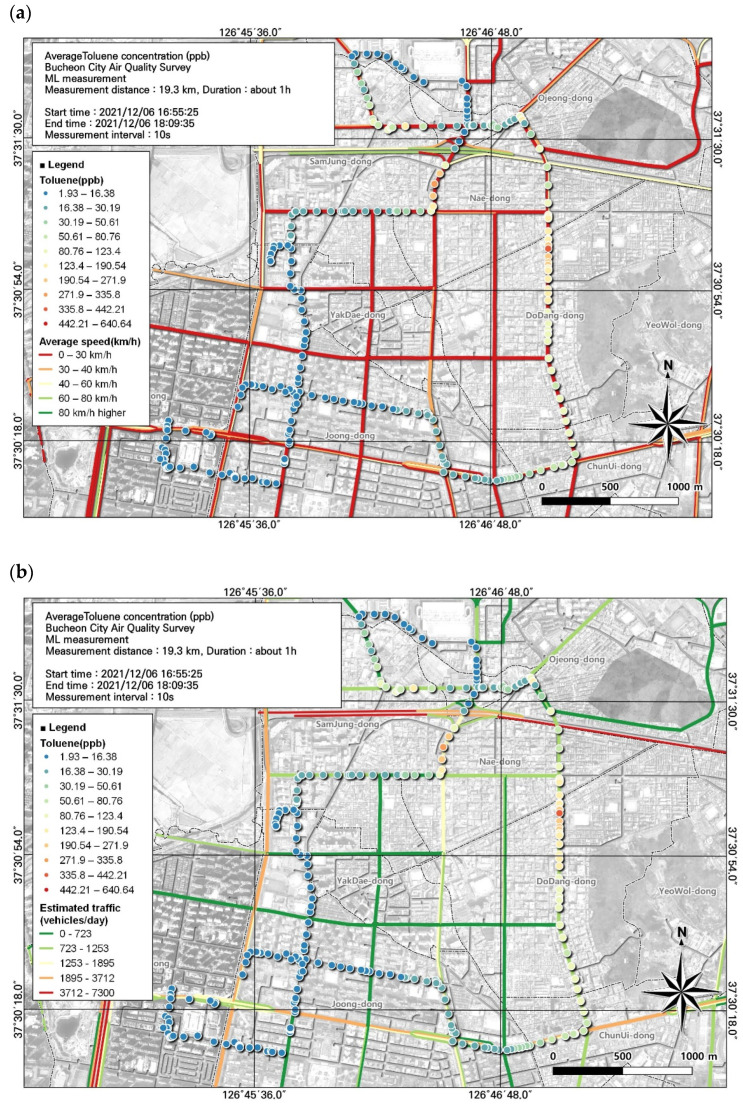
Associations of VOC (toluene) concentrations with (**a**) average traffic speed and (**b**) estimated traffic volume during the morning rush hour, corresponding to Figure 4.

**Figure 10 toxics-12-00511-f010:**
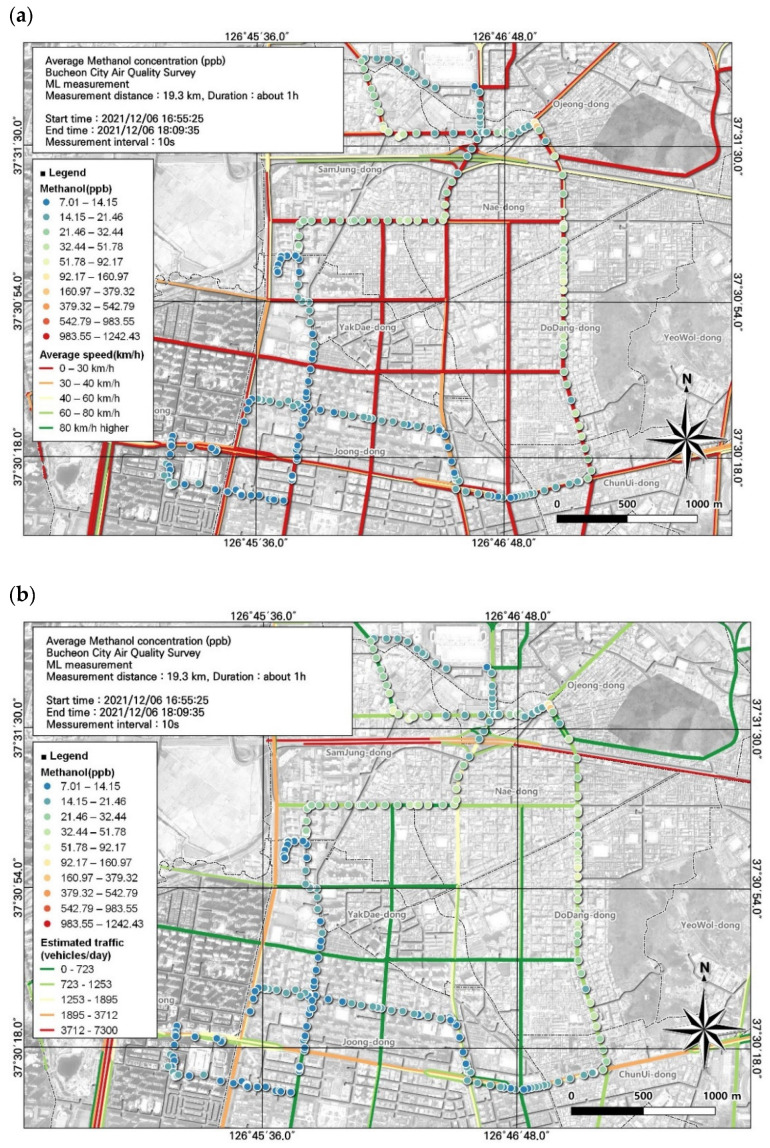
Associations of VOC (methanol) concentrations with (**a**) average traffic speed and (**b**) estimated traffic volume during the morning rush hour, corresponding to Figure 5.

**Figure 11 toxics-12-00511-f011:**
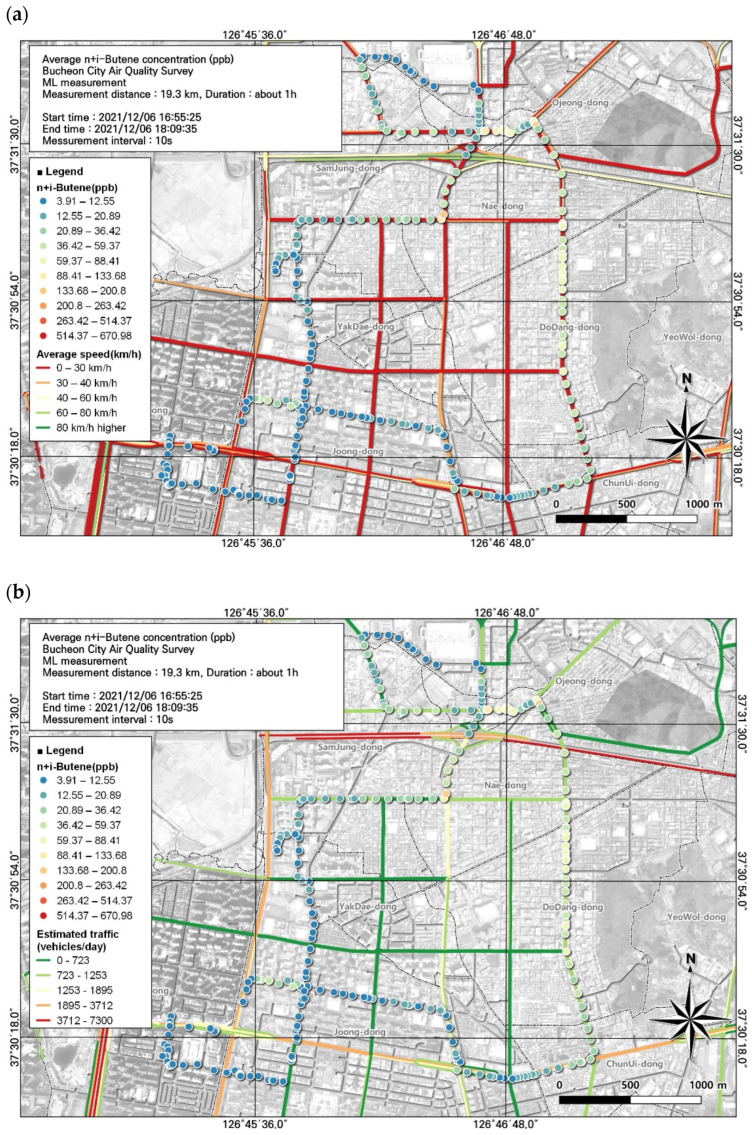
Associations of VOC (n + i-butene) concentrations with (**a**) average traffic speed and (**b**) estimated traffic volume during the morning rush hour, corresponding to Figure 6.

**Figure 12 toxics-12-00511-f012:**
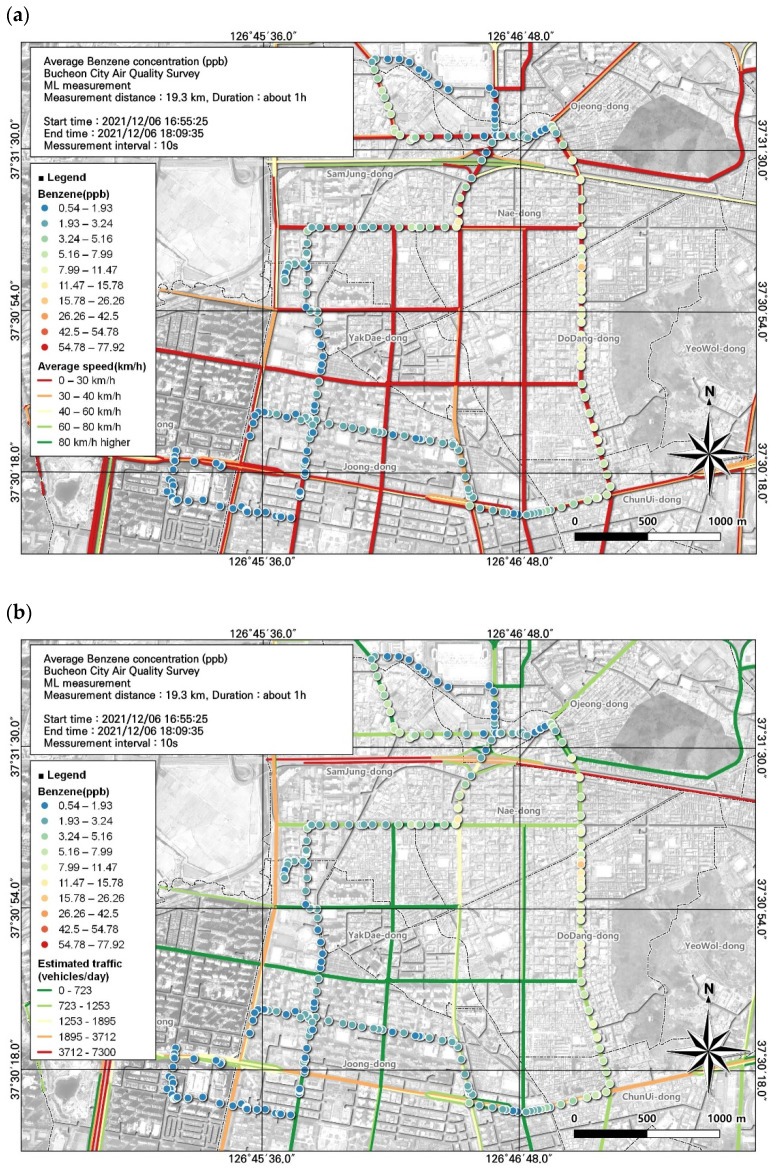
Associations of VOC (benzene) concentrations with (**a**) average traffic speed and (**b**) estimated traffic volume during the morning rush hour, corresponding to Figure 7.

**Table 1 toxics-12-00511-t001:** Measurements collected and instruments used in this study. GPS, Global Positioning System; PTR-ToF-MS, proton-transfer-reaction time-of-flight mass spectrometry; VOC, volatile organic compound.

Measurement	Instrument	Unit	Interval
Latitude, longitude	GPS system	°	1 s
Elevation	GPS system	m	
Speed	GPS system	km/h	
VOCs	PTR-ToF-MS (Ionicon Analytik, Innsbruck, Austria)	ppb	6 s

**Table 2 toxics-12-00511-t002:** VOC targets were analyzed using PRT-ToF-MS.

Compound	Formula	Molar Mass	Protonated Mass	Chemical Abstracts Service
Formaldehyde	CH2O	30.026	31.018	50-00-0
Methanol	CH4O	32.04	33.033	67-56-1
Acetonitrile	C2H3N	41.053	42.034	75-05-8
Propylene	C3H6	42.08	43.054	115-07-1
Propane	C3H8	44.097	44.997	74-98-6
Acetaldehyde	C2H4O	44.053	45.033	75-07-0
Acrylonitrile	C3H3N	53.064	54.034	107-13-1
1,3-Butadiene	C4H6	54.0916	55.054	106-99-0
n + i-Butene	C4H10	58.124	59.086	106-98-9
Acetic acid	C2H4O2	60.052	61.028	64-19-7
i + n-Pentane	C5H12	72.151	73.065	109-66-0
Benzene	C6H6	78.114	79.054	71-43-2
Toluene	C7H8	92.141	93.07	108-88-3
Styrene	C8H8	104.15	105.07	100-42-5
Ethanol + xylene	C8H10	106.168	107.086	100-41-4
Trimethylbenzene	C9H12	120.195	121.101	95-63-6

**Table 3 toxics-12-00511-t003:** Summary of VOC concentrations measured at the stationary observation site.

Pollutant	Mean ± Standard Deviation (ppb)		Percentage
	6 December	7 December	
Formaldehyde	1.3 ± 0.2	1.2 ± 0.2	1.3%
Methanol	17.5 ± 4.1	15.1 ± 2.9	17.0%
Acetonitrile	1.4 ± 0.4	1.3 ± 0.2	1.4%
Propylene	4.5 ± 3.7	3.7 ± 0.9	4.3%
Propane	3.4 ± 0.4	3.1 ± 0.4	3.4%
Acetaldehyde	8.1 ± 1.7	7.6 ± 2.1	8.2%
Acrylonitrile	0.2 ± 0.1	0.2 ± 0.1	0.2%
1,3-Butadiene	2.6 ± 1.6	6.7 ± 10.6	4.8%
n + i-Butene	12.3 ± 6.7	11.3 ± 2.7	12.3%
Acetic acid	9.0 ± 6.8	10.1 ± 3.9	10.0%
i + n-Pentane	8.0 ± 6.3	10.7 ± 4.4	9.7%
Benzene	2.3 ± 0.5	1.8 ± 0.3	2.1%
Toluene	15.1 ± 7.4	16.5 ± 3.9	16.5%
Styrene	1.1 ± 0.4	0.7 ± 0.2	1.0%
Ethanol + xylene	7.9 ± 1.6	5.1 ± 1.3	6.8%
Trimethylbenzene	1.2 ± 0.3	0.8 ± 0.2	1.1%

**Table 4 toxics-12-00511-t004:** Daily concentration of each pollutant measured from the mobile laboratory.

Pollutant	Mean ± Standard Deviation (Unit: ppb)						
	6 December					7 December	
	1st	2nd	3rd	4th	5th	6th	7th
Formaldehyde	1.6 ± 0.3	1.4 ± 0.3	1.2 ± 0.2	1.3 ± 0.2	1.4 ± 0.3	1.3 ± 0.2	1.3 ± 0.2
Methanol	19.7 ± 7.5	19.2 ± 9.6	10.8 ± 2.2	12.2 ± 5.4	34.4 ± 102.7	14.7 ± 5.7	14.9 ± 6.3
Acetonitrile	1.5 ± 0.4	1.3 ± 0.5	0.9 ± 0.2	1.0 ± 0.2	1.5 ± 0.7	1.3 ± 0.3	2.8 ± 6.0
Propylene	5.0 ± 2.2	4.6 ± 3.0	2.2 ± 1.2	3.0 ± 2.0	6.1 ± 6.4	4.0 ± 1.6	16.4 ± 46.4
Propane	4.1 ± 0.5	3.7 ± 0.5	3.1 ± 0.4	3.2 ± 0.4	3.7 ± 0.6	3.4 ± 0.5	3.3 ± 0.4
Acetaldehyde	12.2 ± 3.44	11.9 ± 2.9	7.6 ± 1.7	8.1 ± 1.4	12.4 ± 5.5	8.9 ± 2.3	10.2 ± 1.4
Acrylonitrile	0.2 ± 0.1	0.2 ± 0.1	0.1 ± 0.1	0.1 ± 0.1	0.2 ± 0.1	0.2 ± 0.1	0.2 ± 0.1
1,3-Butadiene	3.3 ± 2.9	2.6 ± 1.5	1.4 ± 0.5	1.8 ± 0.8	2.6 ± 2.0	2.1 ± 0.8	5.1 ± 12.0
n + i-Butene	18.6 ± 22.1	14.4 ± 8.2	10.9 ± 30.0	15.9 ± 21.8	32.6 ± 57.7	15.2 ± 9.5	16.5 ± 12.9
Acetic acid	7.8 ± 8.1	12.0 ± 9.2	6.4 ± 3.3	10.2 ± 8.7	20.4 ± 28.0	9.2 ± 5.2	11.6 ± 8.2
i + n-Pentane	11.3 ± 24.8	9.5 ± 8.0	4.5 ± 2.5	6.6 ± 2.6	11.4 ± 12.5	8.7 ± 4.4	10.8 ± 7.6
Benzene	3.2 ± 2.1	2.7 ± 1.5	1.6 ± 1.1	2.3 ± 1.7	3.8 ± 3.2	2.1 ± 1.6	5.0 ± 10.3
Toluene	28.0 ± 44.5	27.5 ± 22.1	13.4 ± 28.5	21.4 ± 21.6	48.3 ± 67.2	23.6 ± 18.3	45.8 ± 82.1
Styrene	1.2 ± 0.7	1.2 ± 1.5	0.7 ± 0.6	0.9 ± 0.4	5.8 ± 16.9	0.9 ± 0.4	1.2 ± 1.1
Ethanol + xylene	10.8 ± 8.6	10.1 ± 7.1	4.9 ± 3.9	8.0 ± 5.2	15.5 ± 16.3	5.6 ± 4.4	12.1 ± 17.8
Trimethylbenzene	2.1 ± 2.7	1.6 ± 0.9	0.7 ± 0.1	1.4 ± 1.6	2.5 ± 3.2	1.1 ± 0.7	3.2 ± 6.4

**Table 5 toxics-12-00511-t005:** Summary of VOC concentrations measured at the mobile laboratory site.

Pollutant	Mean ± Standard Deviation (ppb)		Percentage
	6 December	7 December	
Formaldehyde	1.4 ± 0.3	1.3 ± 0.2	1.0%
Methanol	19.3 ± 25.5	14.8 ± 6.0	13.3%
Acetonitrile	1.2 ± 0.4	2.0 ± 3.1	1.3%
Propylene	4.2 ± 2.9	10.2 ± 24.0	5.6%
Propane	3.5 ± 0.5	3.3 ± 0.4	2.7%
Acetaldehyde	10.4 ± 3.0	9.5 ± 1.8	7.8%
Acrylonitrile	0.2 ± 0.1	0.2 ± 0.1	0.1%
1,3-Butadiene	2.3 ± 1.6	3.6 ± 6.4	2.3%
n + i-Butene	18.5 ± 28.0	15.8 ± 11.2	13.4%
Acetic acid	11.4 ± 11.5	10.4 ± 6.7	8.5%
i + n-Pentane	8.7 ± 10.1	9.7 ± 6.0	7.2%
Benzene	2.7 ± 1.9	3.6 ± 5.9	2.5%
Toluene	27.7 ± 36.8	34.7 ± 50.2	24.3%
Styrene	2.0 ± 4.0	1.1 ± 0.7	1.2%
Ethanol + xylene	9.9 ± 8.2	8.8 ± 11.1	7.3%
Trimethylbenzene	1.7 ± 1.9	2.1 ± 3.6	1.5%

**Table 6 toxics-12-00511-t006:** Results of VOC concentrations and other environmental measurements during the measurement period.

Number of Measurements	PM_10_ (µg/m^3^)	PM_2.5_ (µg/m^3^)	NO_2_ (ppm)	Toluene (ppb)	Methanol (ppb)	n + i-Butene (ppb)	Benzene (ppb)
1	79.6 ± 19.7	62.7 ± 13.0	0.08 ± 0.03	28.0 ± 44.5	19.7 ± 7.5	18.6 ± 22.1	3.2 ± 2.1
2	69.1 ± 21.6	52.6 ± 11.7	0.07 ± 0.03	27.5 ± 22.1	19.2 ± 9.6	14.4 ± 8.2	2.7 ± 1.5
3	57.1 ± 38.8	39.5 ± 14.2	0.06 ± 0.02	13.4 ± 28.5	10.8 ± 2.2	10.9 ± 30.0	1.6 ± 1.1
4	63.6 ± 34.5	42.9 ± 11.5	0.07 ± 0.05	21.4 ± 21.6	12.2 ± 5.4	15.9 ± 21.8	2.3 ± 1.7
5	91.8 ± 17.6	59.2 ± 8.8	0.08 ± 0.03	48.3 ± 67.2	34.4 ± 102.7	32.6 ± 57.7	3.8 ± 3.2
6	71.2 ± 13.8	59.5 ± 6.8	0.06 ± 0.02	23.6 ± 18.3	14.7 ± 5.7	15.2 ± 9.5	2.1 ± 1.6
7	67.4 ± 25.8	57.6 ± 17.4	0.05 ± 0.02	45.8 ± 82.1	14.9 ± 6.3	16.5 ± 12.9	5.0 ± 10.3

**Table 7 toxics-12-00511-t007:** Correlation analysis between variables for each substance.

Measurements	PM_10_ (µg/m^3^)	PM_2.5_ (µg/m^3^)	NO (ppm)	NO_2_ (ppm)	NOx (ppm)
Toluene	0.34	0.47	0.05	0.03	0.05
Methanol	0.02	−0.01	0.10	0.01	0.09
n + i Butene	0.17	0.13	0.16	−0.01	0.14
Benzene	0.39	0.55	0.08	0.03	0.08

**Table 8 toxics-12-00511-t008:** Correlation analysis between each substance and average speed and expected traffic volume.

Measurements	Average Speed	Expected Traffic Volume
Toluene	0.14	−0.01
Methanol	0.05	−0.02
n + i Butene	0.09	−0.03
Benzene	0.14	0.01

## Data Availability

The original data presented in the study are included in the article; further inquiries can be directed to the corresponding author.

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
