# Peer review of "GIS-Based Analysis of Volatile Organic Compounds in Bucheon, Korea, Using Mobile Laboratory and Proton-Transfer-Reaction Time-of-Flight Mass Spectrometry Methods"

_toxics, 2024, doi:10.3390/toxics12070511_

Round 1

Reviewer 1 Report (New Reviewer)

Comments and Suggestions for Authors

The manuscript presents a GIS-based anaysis of volatile organic compounds in Boucheon, Korea using a mobile laboratory, in order to decipher the spatiotemporal variability within areas of different characteristics. Meteorological parameters and traffic data were combined along with air quality parameters in order to correlate specific sources with respective VOCs. The highest levels were recorded for toluene, methanol and n+i butene, and were identified close to the industrial zone of Ojeong. 

Although the study is very interesting, there are some issues to be addressed before moving on with publication.

Major issues:

1) More details should be provided concerning the mobile measurements. How long is the inlet, what is the residence time, was the line heated? What about wall losses? How often were the calibrations?

2) Although very interesting and containing all the information, in all spatial figures the concentration points along the routes are barely visible.. I would suggest either enlarging the points or use other colors that do not blend in with the background map.

3) The correlation analysis interpretation appears a bit farfetched. How can we talk about "relatively high correlation" when the correlation coefficient is at 0.09 (e.g. methanol vs NOx Table 7). I would rephrase to "the highest amongst the observed observed values" or something similar. Similarly for the others which are presented as high, but are very low.

4) Since acetonitrile results are also presented, which is a marker for biomass burning, and the observed values are quite high (~1.3 ppb, and December 7th the 7th measurement at 2.8+-6!!) I would also expect a discussion about this as well, since the route also included residential areas and the measurements took place during winter. To my opinion this would be interesting. 

5) The fact that for both benzene and PM2.5 the recorded values are quite high should be also put into context with the annual limits. Maybe also plan a campaign during summer or a few days each month to establish a seasonal variability? 

Technical corrections:

Title 2.2.1 Laboratory equipment (capitalize)

In Figure 1 I would also add the stationary site, or is it the one mentioned as AQMS? Please clarify and make a bit larger the marker.

L519: What does this sentence mean "GIS-based analysis was conducted on the impact of etc."??

Author Response

Reviewer 2 Report (New Reviewer)

Comments and Suggestions for Authors

The manuscript deals with a GIS analysis of VOCs in Bucheon (Korea), analyzed by a Mobile Laboratory using TOF mass spectrometry. The VOCs concentrations have been measured in different contexts including industrial and residential areas of Bucheon city. Pollution sources have been also analyzed as traffic, road level, traffic volume, average speed of air pollution. The manuscript is well-organized and the results are promising for replicated campaigns. Some Minor Revisions are suggested:

- VOCs have been measured by Mobile Laboratory as high level during winter on December. Any data of VOCs during summer or spring to compare pollution VOCs level as low as expected during warmer seasons.

- additional recent references should be added in the list.

After these Minor Revisions, the manuscript should be publishable.

Comments on the Quality of English Language

Minor editing of Emglish is suggested.

Round 2

Reviewer 1 Report (New Reviewer)

Comments and Suggestions for Authors

The authors adequately addressed all the issues raised, and especially all the figures are now more readable. Therefore I do not see any reason not to move on with the publication of the manuscript.

Reviewer 2 Report (New Reviewer)

Comments and Suggestions for Authors

The revised manuscript is enhanced by inputs of the referees. Now, it can be published and suitable to journal.

Comments on the Quality of English Language

Minor editing of English should be beneficial for manuscript.

This manuscript is a resubmission of an earlier submission. The following is a list of the peer review reports and author responses from that submission.

Round 1

Reviewer 1 Report

Comments and Suggestions for Authors

This revised manuscript more or less addressed my general questions from the last round. The result discussion improved. The results from this study could be useful and could serve as the base for future studies on VOCs in the region. However, I still have questions for the authors to address before the manuscript can be published.

General question:

In section 3, you have both results and discussion, so I would suggest that you could shorten section 4 by highlighting only the key findings and implications, and moving the discussion on details to section 3. So the manuscript can be more clear and concise.

Questions on details:

Line 98: Is haze typical in that region in winter? This could be very interesting if haze is not usual there in winter. Do you have any aerosol measurements on those two days, such as mass concentration or even speciation in that region and in winter?

Line 99: “fine dust”, do you mean fine particles (or aerosols)?  I would suggest changing the wording, since dust particles are usually coarse particles. 

Line 102: I suggest changing “moving measurements” into “mobile measurements” to be consistent.

Comments on the Quality of English Language

I would suggest proof reading the whole text again to avoid half-sentences and other errors.

Reviewer 2 Report

Comments and Suggestions for Authors

Thank you very much for your efforts. In my oppinion, the manuscript was improved, but the most critical points were left undone or answered in a very superficial way. 

In addition to the short evaluation period, data analysis in terms of correlation between concentrations and other environmental factors remains scientifically unsound. The inclusion of so many graphs makes reading difficult and it would be interesting to present a more systematic and summarized analysis of all the data/information.

Thus, the manuscript needs to be reformulated in terms of data analysis capable of extracting more useful and trustworthy information.

Comments on the Quality of English Language

Minor editing of English language required

Round 2

Reviewer 1 Report

Comments and Suggestions for Authors

This improved version is good for publishing.

Reviewer 2 Report

Comments and Suggestions for Authors

In general terms, the authors did not understand or did not want to understand my comments. The main flaws of the initial version continue to remain in this latest version. I tried to get the authors to reflect and be able to conceive something with much more scientific value, but they rejected this possibility.

However, as I recognize that the study presents VOC data with high temporal and spatial detail, due to the use of expensive technologies that are not within the reach of many research teams.